

# Unveiling atmospheric transport and mixing mechanisms of ice nucleating particles over the Alps

Jörg Wieder[1], Claudia Mignani[2], Mario Schär[1], Lucie Roth[1], Michael Sprenger[1], Jan Henneberger[1], Ulrike Lohmann[1], Cyril Brunner[1], and Zamin A. Kanji[1]

[1]ETH Zurich, Institute for Atmospheric and Climate Science, Zurich, Switzerland
[2]Department of Environmental Sciences, University of Basel, Basel, Switzerland

**Correspondence:** Jörg Wieder (joerg.wieder@env.ethz.ch), and Zamin A. Kanji (zamin.kanji@env.ethz.ch)

**Abstract.** Precipitation over the mid-latitudes originates mostly from the ice phase within mixed-phase clouds, signifying the importance of initial ice crystal formation. Primary ice crystals are formed on ice nucleating particles (INPs), which are sparsely populated in the troposphere. INPs are emitted by a large number of ground-based sources into the atmosphere, from where they can get lifted up to cloud heights. Therefore, it is vital to understand vertical INP transport mechanisms, which are particularly complex over orographic terrain. We investigate the vertical transport and mixing mechanisms of INPs over orographic terrain during cloudy conditions by simultaneous measurements of in situ INP concentration at a high valley and a mountaintop site in the Swiss Alps in late winter 2019. On the mountaintop, the INP concentrations were on average lower than in the high valley. However, a diurnal cycle in INP concentrations was observed at the mountaintop, which was absent in the high valley. The median mountaintop INP concentration equilibrated to the concentration found in the high valley towards the night. We found that in nearly 70% of the observed cases INP-rich air masses were orographically lifted from low elevation upstream of the measurement site. In addition, we present evidence that over the course of the day air masses containing high INP concentrations were advected from the Swiss plateau towards the measurement sites, contributing to the diurnal cycle of INPs. Our results the local INP concentration enhancement over the Alps during cloud events.

## 1 Introduction

Precipitation serves as a major source of fresh water in the global hydrological cycle (Field and Heymsfield, 2015). Changing distribution patterns in precipitation may increase the risk of droughts and floods (Rosenfeld et al., 2008; Chow et al., 2013), and could be associated with political conflicts (Tignino, 2010). Weather forecasts and climate change projections of precipitation remain difficult for many reasons, including the lack of available data and an understanding of relevant cloud processes (Hegerl et al., 2015). The formation of precipitation is linked to the evolution of cloud microphysical properties which is particularly complex for *mixed-phase clouds* (MPCs) due to the co-occurrence of supercooled liquid droplets and ice crystals (e.g. Wegener, 1911; Pruppacher and Klett, 2010; Lohmann et al., 2016b). MPCs can exist in the temperature range between 0 °C and





approximately $-38$ °C, thus, they are frequently observed over the Alps (e.g., Henneberger et al., 2013; Lohmann et al., 2016a; Beck et al., 2017). Atmospheric aerosol particles can alter the cloud microphysics, subsequently affecting precipitation

formation and evolution (e.g. Borys et al., 2003; Rosenfeld et al., 2008; Muhlbauer and Lohmann, 2009), for which the ice phase is important (Field and Heymsfield, 2015; Mülmenstädt et al., 2015; Heymsfield et al., 2020). In the absence of external snow sources e.g., snow sedimenting from higher cloud layers (*seeder-feeder effect*, see e.g. Ramelli et al. (2021a); Lee et al. (2000) and references therein), ice crystals in MPCs are initially formed via heterogeneous ice nucleation. Heterogeneous ice nucleation is catalyzed by sparse aerosol particles called *ice nucleating particles* (INPs) by providing a surface for nucleation

(see e.g. Wegener, 1911; Vali, 1971; Pruppacher and Klett, 2010; Murray et al., 2012; Vali et al., 2015). The aerosol effect on the cloud microstructure highlights the crucial need to understand the spatiotemporal INP availability - specially the vertical transport to cloud heights. In the atmosphere, a variety of aerosol particles has been found to act as INPs, such as mineral dust particles, pollen, biological or organic compounds, and also anthropogenic created particles from e.g., biomass burning or combustion (Kanji et al., 2017; Huang et al., 2021, and references therein). Depending on the source, these particles have

a different ice nucleation temperature (Kanji et al., 2017; Huang et al., 2021, and references therein). However, the spatial abundance and distribution of INPs in the atmosphere is highly uncertain (see e.g. Demott et al., 2010; Kanji et al., 2017; Murray et al., 2021).

Most atmospheric aerosol particles are found within the first kilometers above ground in the troposphere, i.e. are confined within the *planetary boundary layer* (PBL). The PBL is in a well mixed state resulting in a constant potential temperature

and aerosol number concentrations with height (see e.g. Stull, 1988; Chow et al., 2013). The top of the PBL is, among other indicators, characterized by an abrupt decrease in aerosol number concentration (Stull, 1988; Chow et al., 2013). In contrast to the idealized description of the PBL over a flat surface, an accurate description of the PBL over complex mountainous terrain is complicated by a variety of processes such as, orographic gravity waves, moist convection, and turbulent transport (see e.g., Rotach and Zardi, 2007; Lehner and Rotach, 2018). Whereas the transport of pollutants (aerosol particles or gas) has

been previously studied (e.g. Baumann et al., 2001) less is known about the transport of INP. With the PBL being confined close to the surface, higher aerosol number concentrations can be found at the same altitude (m a.s.l.) over orographic terrain compared to, for example, flat lands at sea level. This raises the question if an increased availability of INPs at higher altitudes in orographic regions is a common feature.

Conen et al. (2017) reported an altitudinal gradient of daily-averaged concentrations of INPs active at $-8$ °C based on

parallel measurements done between May and September at three sites of different altitudes located in Switzerland and found a decrease of roughly 50% with each kilometer in altitude. Poltera et al. (2017) found that near the High Altitude Research Station Jungfraujoch (3580 m a.s.l.) the vertical extent of the PBL can be increased by convection over the course of a day. Lacher et al. (2018) observed that INP concentrations around $-31$ °C are elevated during times of boundary layer intrusions at the Jungfraujoch, suggesting that INPs are also more numerous within the PBL aerosol. Relating INP concentration as an aerosol

subset to the ambient aerosol concentration is highly dependent on the air mass and precipitation history (Mignani et al., 2021), as aerosol particles and INPs follow different accumulation and depletion mechanisms. While both (arbitrary) aerosol particles and INPs can potentially be removed by dry or wet deposition, INPs can additionally be removed if embedded in hydrometeors





after acting as INPs. Furthermore, INPs can be added due to resuspension during and after rainfall (e.g., Huffman et al., 2013; Seifried et al., 2021). The INP concentration within a plume of freshly emitted mineral dust aerosol can be deduced from the aerosol properties (using parameterizations like Niemand et al., 2012; Demott et al., 2015). Upon precipitation formation the relation between INPs and aerosol may no longer hold (Mignani et al., 2021), as the sources and sinks of INPs and aerosol particles are disproportional to each other. Thus, transport mechanisms to higher altitude that replenish the INP concentration after cloud formation are one crucial aspect we study here to understand the variations in INP concentration.

Among many other overviews, Chow et al. (2013) and Wekker and Kossmann (2015) summarized a vast variety of wind systems in orographic terrain. Winds can increase the PBL height, which would otherwise be confined to the valley, due to induced vertical motion of the valley atmosphere. A specific wind system can be seen as a superposition of three drivers, namely (i) the synoptic wind speed and direction, (ii) the diurnal (thermally driven) mountain-valley breeze, and (iii) the vertical stability of the air masses within the valley. Combining the three drivers to varying degree and considering an elongated valley, three generalized wind systems are imaginable: First, the synoptic flow arriving at the orographic barrier pushes the air masses near the surface towards the mountain ridges and tops (*orographic lifting* as a result of *advection*). In the absence of upslope winds and given weak stratification of the air masses leeward of the ridge, the air masses will be further pushed down the leeward valley. This synoptic forcing can result in a similar effect as the diurnal warming of the slopes, yet it is consequence of the flow being deflected by the mountain topography (Chow et al., 2013). Second, especially on sunny days, air close to the ground of slopes on the sun exposed side of a valley heats up faster than the surrounding air, causing an upslope wind that is confined to a few hundred meters above the slopes. This situation reverts overnight when the slopes cool down, creating a downslope wind (Schmidli and Rotunno, 2010; Wekker and Kossmann, 2015). It is required that the vertical stability allows for the vertical movement of air masses and that the synoptic wind is not too strong deflecting or fully eliminating the slope winds (Barry and Chorley, 2003). This wind system is prominent during summer time and weaker in winter time, as snow cover on slopes causes downslope winds due to the cooling of air masses (*katabatic winds*) counteracting the upslope wind. Mott et al. (2015) found that an increasing fraction of snow covered slope area resulted in stronger katabatic winds. This effect could reduce the strength of daytime upslope winds in winter. A third option is created by the meso-scale wind flowing over the orographic barrier. If the wind direction is perpendicular to the valley axis and its speed is strong enough, it could cause large-scale eddies, mixing the atmosphere within the valley (*rotors*, see Chow et al., 2013, chapter 3.3).

In this study, we investigate how the distribution of INPs between a mountain and a valley site in the Alps changes spatiotemporally and discuss the underlying transport mechanisms. Whereas in situ INP concentrations were often reported either in flat regions (e.g., Mason et al., 2016; Chen et al., 2018; Ladino et al., 2019; Paramonov et al., 2020) or at mountaintops (e.g., DeMott et al., 2003; Richardson et al., 2007; Klein et al., 2010; Stopelli et al., 2014; Conen et al., 2015; Boose et al., 2016a, b; Lacher et al., 2017; Creamean et al., 2019; David et al., 2019; Mignani et al., 2021; Brunner and Kanji, 2021), we observed in situ INP concentrations simultaneously on a mountaintop and a nearby valley site (1060 m height difference, 3.65 km horizontal distance) in a complex terrain. Measurements were conducted over eight weeks at high sampling frequency during cloud events (for 20 minutes every 1.5 hour) in late winter 2019 (February and March). This allowed for drawing conclusions about the vertical distribution and mixing based on two extensive datasets of INP concentrations at both sites. We analyzed the distri-





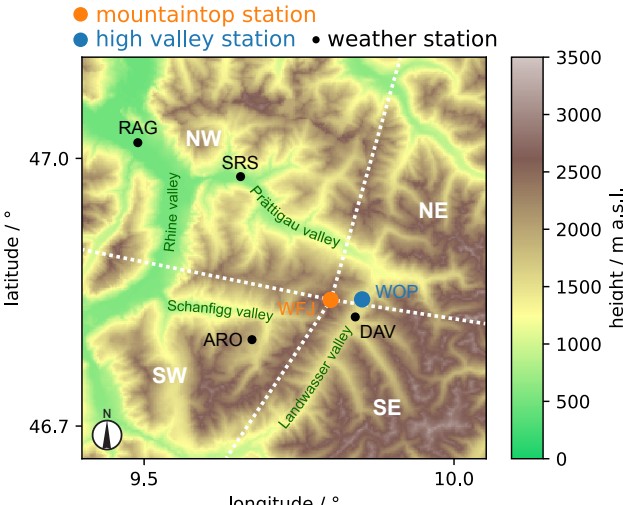

**Figure 1.** Locations of the measurement sites. The main aerosol measurements were located at Wolfgangpass (blue, WOP, 1631 m a.s.l.) and Weissfluhjoch (orange, WFJ, 2693 m a.s.l.). The horizontal distance between WOP and WFJ measures 3.65 km. Meteorological data was retrieved from measurement stations in Arosa (ARO, 1880 m a.s.l.), Bad Ragaz (RAG, 498 m a.s.l.), Davos Dorf (DAV, 1598 m a.s.l.), Schiers (SRS, 628 m a.s.l.), and at Weissfluhjoch (WFJ, 2693 m a.s.l.). The white dashed lines indicate the edges of the four meso-scale wind direction sectors (NW, NE, SW, SE) used for the analysis based on the topography used in Section 3.3 - 3.5. The topography was extracted from the digital height model DHM200 from the Federal Office of Topography swisstopo.

bution of INPs at temperatures above $-20\,°C$ relevant for MPCs. We present the difference in INP concentration between both sites and show evidence of a diurnal cycle in INP concentrations measured on the mountaintop. Based on the local topography,

we describe how dynamic transport processes can enhance the INP concentration at the mountaintop site.

## 2   Measurement setup

During the RACLETS (Role of Aerosols and CLouds Enhanced by Topography on Snow) campaign in the region of Davos, Switzerland, in February and March 2019, a complementary set of aerosol, cloud, precipitation and snow measurements was collected (Envidat, 2019; Walter et al., 2020; Mignani et al., 2021; Ramelli et al., 2021b, a; Lauber et al., 2021; Georgakaki

et al., 2021). Two similarly equipped aerosol measurement sites, one located on a saddle at the entrance of a high valley (Wolfgangpass, 1631 m a.s.l., hereafter referred to as WOP) and the other on a mountaintop (Weissfluhjoch, 2693 m a.s.l., hereafter referred to as WFJ) are shown in Figure 1.

### 2.1   Aerosol and INP measurements

Ambient aerosol was sampled through a heated total inlet at both WFJ and WOP and analyzed for properties including size

and ice nucleation activity. At WOP, the inlet was mounted on a measurement trailer and consisted of a 1.5 m long vertical





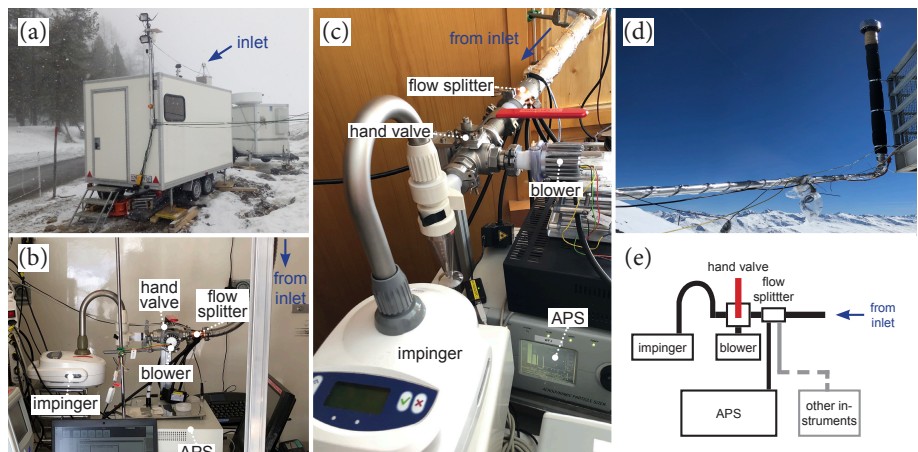

**Figure 2.** Aerosol measurement setups at WOP and WFJ. The aerosol trailer at WOP (a) with the heated inlet on the roof (blue arrow), the setup inside the aerosol trailer (b), the setup at WFJ (c) with the inlet pipe indicated (blue arrow) and the inlet mounted on a railing at WFJ (d). The schematic (e) depicts the flow connections used at both sites.

pipe (25 mm diameter) and a matching 0.5 m long 90°-bend (Figure 2a and 2b). At WFJ, the inlet with a diameter of 50 mm consisted of a 2.5 m vertical pipe and a 4.5 m long inclined pipe (height loss of approx. 0.8 m across total length, see Figures 2c and 2d). Both inlets were capped with a hat and all outside parts (including the hat, at WOP approx. first 0.7 m, at WFJ all) were heated to 40 °C. Contributions of resuspended particles from the snow-covered surface around the measurement sites cannot fully be excluded but are unlikely to have added significantly to the sampled aerosol due to the inlet's design (Mignani
et al., 2021).

Downstream of the inlet, a custom-made flow splitter was mounted followed by a three-way ball valve (Model 120VKD025-L, Pfeiffer Vacuum, Germany) connecting a blower (Model U71HL, Micronel AG, Switzerland) and a high flow-rate impinger (Coriolis® µ, Bertin Instruments, France), both operating at 300 L min$^{-1}$ (see Figure 2b for WOP and Figure 2c for WFJ).
The impinger collected particles larger than 0.5 µm (collection efficiency of 50%, 80% and 94% for 0.5 µm, 2 µm and 5 µm particles respectively, personal communication with manufacturer, July 3 2020). During times when the impinger was idle, the ball valve was switched to the blower to create a make up flow, thereby guaranteeing a constant flow through the inlet at all times. From the splitter, smaller connections with diameters of approximately 5 mm branched off at a 45° angle around the center axis. At the lowest sampling line, a commercial Aerodynamic Particle Sizer Spectrometer (APS, Model 3321, TSI
Corp., US) recorded size distributions continuously (see Figure 2e). Aerodynamic diameter was converted to physical diameter using a shape factor $\chi = 1.2$ and an assumed particle density $\rho = 2 \, \mathrm{g \, cm^{-3}}$ (Thomas and Charvet, 2017).

Using the aerosol-to-liquid impingers, air samples were collected for analysis of ambient INP concentrations at both sites (as also already described for WFJ in Mignani et al., 2021). The standard protocol consisted of sampling air for 20 minutes (corresponding to 6 m$^3$ of air) in 15 mL of ultra-pure water (W4502-1L, Sigma-Aldrich, US). To compensate for evaporational



loss while operation of the impinger, the sample liquid was refilled to 15 mL after 10 and 20 minutes. The sampling was

directly followed by analysis for INP concentration on site using the drop-freezing instrument DRINCZ (David et al., 2019) at

WOP and LINDA (Mignani et al., 2021; Stopelli et al., 2014) at WFJ. For both instruments, sample liquid was pipetted out into

small aliquots ($96 \times 50\ \mu\mathrm{L}$ for DRINCZ, $52 \times 100\ \mu\mathrm{L}$ for LINDA) and cooled down in a cryostat until all aliquots froze. While

cooling, pictures of the droplet arrays were obtained with a digital camera mounted above the droplet assay (every 15 seconds

and 5 seconds for DRINCZ and LINDA, respectively) to detect freezing of individual aliquots with respect to temperature in

the post-processing. Cumulative INP concentrations at integer temperatures $T$ were determined according to Vali (1971, 2019)

as

$$n_{\mathrm{INP}}(T) = -\frac{\ln[1 - \mathrm{FF}(T)]}{V_{\mathrm{a}} \cdot C} \tag{1}$$

where $\mathrm{FF}(T)$ is the fraction of frozen wells at temperature $T$, and $V_{\mathrm{a}}$ is the volume of an individual aliquot. $C$ is a normalization

factor in order to calculate the INP concentration per standard liter of sampled air ($\mathrm{StdL}^{-1}$) and is defined as

$$C = \frac{F_{\mathrm{Coriolis}} \cdot t_{\mathrm{sample}}}{V_{\mathrm{Coriolis}}} \cdot C_{\mathrm{StdL}}\,, \quad \text{where} \quad C_{\mathrm{StdL}} = \frac{p_{\mathrm{ambient}}}{p_{\mathrm{ref}}} \cdot \frac{T_{\mathrm{ref}}}{T_{\mathrm{ambient}}} \tag{2}$$

with $F_{\mathrm{Coriolis}}$ being the impinger flow rate ($300\ \mathrm{L\,min^{-1}}$), $t_{\mathrm{sample}}$ the sampling time (20 minutes), $V_{\mathrm{Coriolis}}$ the sample volume

after sampling (15 mL), $p_{\mathrm{ambient}}$ and $T_{\mathrm{ambient}}$ the mean ambient pressure and temperature during sampling using the reference

of a standard liter with $p_{\mathrm{ref}}$ = 1013.25 hPa and $T_{\mathrm{ref}}$ = 273.15 K. INP concentrations have been corrected for the background

of the blank ultra-pure water as described in David et al. (2019). In rare cases of very active samples (i.e. all droplets froze

already above $-10\ °\mathrm{C}$), dilutions of the sample were prepared and analysed. In this study one value for INP concentration per

temperature and sample was desired, therefore, samples and their dilutions were combined by using the number of droplets

freezing at each temperature step as proportional weights. For a sample $a$ and its dilution $b$, the combined INP concentration

$n_{\mathrm{INP,comb}}(T)$ at temperature $T$ was calculated as

$$n_{\mathrm{INP,comb}}(T) = \sum_{m=0}^{T} \frac{\Delta N_{\mathrm{a}}(m) \cdot k_{\mathrm{a}}(m) + \Delta N_{\mathrm{b}}(m) \cdot k_{\mathrm{b}}(m)}{\Delta N_{\mathrm{a}}(m) + \Delta N_{\mathrm{b}}(m)} \cdot \Delta T \tag{3}$$

where $\Delta N_{\mathrm{a/b}}(m)$ are the number of frozen droplets for sample $a$ and $b$, respectively, at temperature $m$, $k_{\mathrm{a/b}}(m)$ are the dif-

ferential INP concentrations for sample $a$ and $b$, respectively, at temperature $m$ and $\Delta T$ is the temperature bin size (definitions

of the variables analogous to Vali, 2019).

In this study we will compare the INP concentrations measured at the two aerosol measurement sites. To ensure the compa-

rability of both instruments, a comparison study was conducted. The results from an ambient aerosol sample collected with the

impinger used in this study and analyzed with DRINCZ and LINDA is published in Miller et al. (2021, Figure 4b). We found

agreement within a factor of 2 for temperatures below $-8\ °\mathrm{C}$ and within a factor of 5 above $-8\ °\mathrm{C}$ with LINDA reporting the

higher INP concentrations in both temperature regimes. The observed larger deviation at warmer temperatures is not necessar-

ily only attributed to instrumental differences, but also to the large uncertainty arising from the sparsity of INPs with increasing

temperatures.





### 2.1.1 INP measurement strategy and data selection

The RACLETS campaign aimed to better understand MPCs and their precipitation formation over the Alps. To this end, the sampling of ambient air for consequent INP analysis was targeted during cloud events and was synchronized for most samples between the two sites (WFJ and WOP). The first sample was taken around 2 hours before and the last 2 hours after the presence
of a cloud over the sampling region. Within this time frame, samples were taken at 1.5 - 2 hour intervals. The analysis focuses on the diurnal cycle of INP concentrations at both sites and categorizes the samples into three periods of the day (morning (03:00-11:59 UTC), afternoon (12:00-17:59 UTC) and night (18:00-02:59 UTC)). Initially, a categorization of four periods (every six hours) was intended, but as there were only five samples at WFJ and two samples at WOP available between 00:00 - 05:59 UTC, this period was split and joined with the adjacent ones. To investigate the diurnal variation in INP concentration,
only sequences of samples were considered when there was at least one sample in three consecutive periods. The samples reported by Mignani et al. (2021) taken at WFJ during a Saharan dust event are excluded from this study to avoid bias from one dominating aerosol type. Applying these criteria to the datasets resulted in 111 samples obtained at each WOP and WFJ (out of a total number of samples of 157 and 155 at WOP and WFJ, respectively).

### 2.2 Meteorological measurements

Meteorological information in Arosa (ARO, 1880 m a.s.l.), Bad Ragaz (RAG, 498 m a.s.l.), Davos Dorf (DAV, 1598 m a.s.l.), Schiers (SRS, 628 m a.s.l.), and at WFJ (2693 m a.s.l.), were retrieved from the corresponding observation stations of the Swiss national weather service MeteoSchweiz (see Figure 1). During the campaign, a weather station installed at WOP monitored the ambient pressure, relative humidity, and temperature. Because the wind speed and direction was not measured at WOP, wind data from the nearest MeteoSchweiz observation station in Davos were used assuming it to be representative for WOP
(in accordance with Georgakaki et al., 2021). The WFJ wind direction was found to agree with the meso-scale wind direction over Davos. Thus, references to the meso-scale wind direction in this study are based on the wind measurements at WFJ.

### 2.3 Back-trajectories

For every full hour of the day, kinematic back-trajectories were calculated for both sites using the Lagrangian analysis tool LAGRANTO (Sprenger and Wernli, 2015; Wernli and Davies, 1997) each extending five days back in time at ten minute
resolution. The back-trajectories started over the Swiss Alps, i.e., a region with complex topography, and therefore could only reliably be calculated if the 3D wind fields were available at sufficient temporal and spatial resolution. To achieve this, we relied on the operational analysis of MeteoSchweiz that builds on the non-hydrostatic COSMO (Consortium for Small-scale Modeling, Baldauf et al. (2011); Schättler et al. (2015)) model with hourly wind fields at a 1-km horizontal resolution. When a trajectory left the COSMO domain (longitude: $0.3° - 17.0°$, latitude: $42.6° - 50.3°$), the calculation continued with the
(coarser) wind fields of the ECMWF operational analysis, at a six-hourly temporal resolution and interpolated to a $1° \times 1°$ latitude/longitude grid. Using the back-trajectories, aerosol footprint maps were generated for the two sites. The temporally closest trajectory to each INP sample was selected and every geographic location (longitude and latitude) was noted for which



the trajectory was less than 500 m above ground indicating a potential aerosol source from these locations. In addition, locations with surface height exceeding 2500 m a.s.l. were omitted as at these locations trajectories passed close to mountaintops where

aerosol sources were excluded.

## 3    Results and discussion

### 3.1    Overview of INP concentrations at WFJ (mountaintop) and WOP (high valley)

The distribution of INPs as a function of temperature of all samples is presented in Figure 3. We found that the across the temperature spectrum median INP concentrations at WOP were approximately three times higher than at WFJ. This supports

the hypothesis of INP concentrations being generally lower at higher altitudes. Conen et al. (2017) reported INP concentrations at $-8\,°C$ on Jungfraujoch (3580 m a.s.l.) being approximately four times lower than at Chaumont (1136 m a.s.l.). Based on their results, the INP concentration at $-8\,°C$ at WFJ are expected to be a factor of roughly 2 lower than at WOP. During our campaign we observed the median INP concentration at $-8\,°C$ to decrease by a factor of 2.4 from WOP to WFJ. Since WOP is located in a high valley and not on a mountain range as Chaumont, a higher decrease is reasonable as aerosol accumulates

in the valley. Moreover, the interquartile ranges of the INP concentrations at WFJ are larger than at WOP throughout the temperature spectrum, representing a larger variability of INP at WFJ (Figure 3). We attribute the observed larger variability at WFJ to the fact that a given perturbation in INP number concentrations leads to a larger relative variability at lower median INP concentrations than at higher median INP concentrations. This suggests that WFJ is more susceptible to aerosol perturbations. For WFJ being located in a ski resort, local aerosol sources, such as cooking in restaurants, smoking tourists and the preparation

of the slopes at night (soot emissions from snow cats) are possible. Generally, small particles from combustion processes can be excluded due to the impinger's 50% collection cut-off size of $0.5\,µm$. In addition, for the investigated temperatures and freezing mode (immersion) the contributions of local point soot emissions to ice nucleation should be negligible (Kanji et al., 2020; Vergara-Temprado et al., 2018; Mahrt et al., 2018; Chou et al., 2013). Furthermore, INP concentrations were not influenced during susceptible periods, e.g. for wind directions from the nearby kitchen to the sampling site (distance $\sim$100 m) during

10 am - 1 pm local time (see Figure A1). Hence, fluctuations in the INP concentration at WFJ are expected to be caused by advected aerosol particles from the surrounding valleys or by long-range transported aerosol. This raises the question if there are times when the INP concentration at WFJ reach (or even exceed) the INPs concentration at WOP. Subsequently, we investigate the diurnal concentrations of INP at both sites in the following section.

### 3.2    Diurnal cycle of INP concentrations

Over orographic terrain, region-specific wind systems (e.g. mountain-valley breeze) can occur in diurnal cycles (see e.g. Nyeki et al., 2000; Chow et al., 2013; Wekker and Kossmann, 2015). A common observation is the increase of the PBL height over the day and its decrease during the night (e.g. Wekker and Kossmann, 2015; Poltera et al., 2017), which also redistributes the aerosols. To investigate the presence of a diurnal cycle in the INP concentration, samples taken at WOP and WFJ are



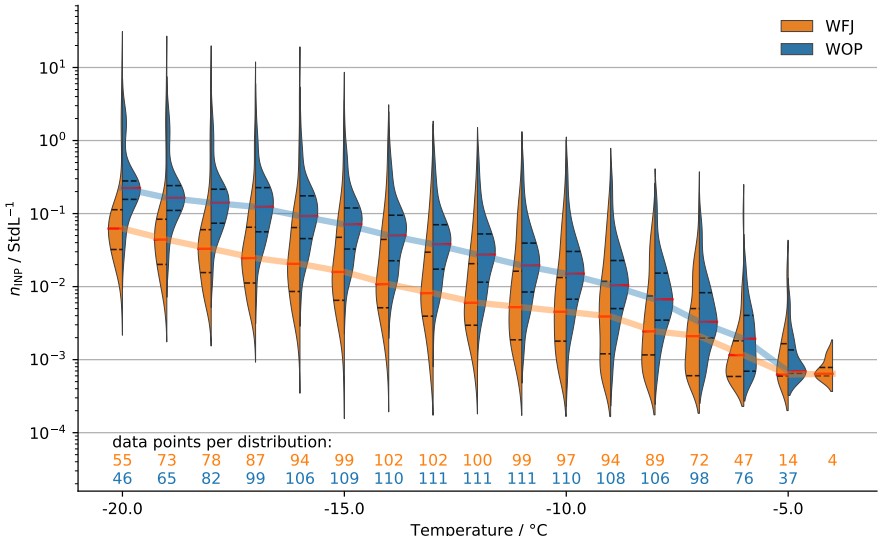

**Figure 3.** Cumulative INP concentration ($n_{\mathrm{INP}}$) violin plots per temperature of all collected samples at WFJ (orange) and WOP (blue). For each distribution the median is given in solid red as well as the 25[th] and 75[th] percentile in dashed black. The lines connect the distribution medians. At the bottom of the figure, the number of data points at each temperature is shown in orange and blue for WFJ and WOP, respectively.

categorized into morning, afternoon, and night (see Section 2.1.1). In Figure 4a, INP concentrations at WOP against INP
concentrations measured in parallel at WFJ are compared. A diurnal pattern is visible across all measurements. In the morning, nearly 90% of the observed INP concentrations at WOP are higher than at WFJ at all temperatures. Over the course of the day, the INP concentrations at WFJ progressively increase reaching similar levels to WOP during the night. To investigate this observed diurnal pattern in detail, we present the median INP concentration between $-8\,^{\circ}\mathrm{C}$ and $-20\,^{\circ}\mathrm{C}$ binned per two hours as a function of the time of day in Figures 4b and 4c. While at WOP median INP concentrations do not exhibit a large variation
over the course of a day, median INP concentrations at WFJ are lowest around 11 UTC and increase by nearly one order of magnitude until 19 UTC. The increase is less pronounced in INP concentrations active at temperatures colder than $-15\,^{\circ}\mathrm{C}$. This could be explained by a paucity in data for colder temperatures where active samples were not included in the analysis, as all their droplets were already frozen above $-15\,^{\circ}\mathrm{C}$. This is shown by a decrease in the number of data points (see Figure 3) towards colder temperatures.
In Figure 5a and 5b the median INP concentration and the median activated fraction ($\mathrm{AF}_{0.5}$, ratio of INP concentration and total aerosol concentration of particles with diameter larger $0.5\,\mu\mathrm{m}$), respectively are shown as a function of temperature for the three time periods (morning, afternoon, night). As there is an increase in median $\mathrm{AF}_{0.5}$ at WOP from the morning to the afternoon, whereas the corresponding median INP concentration remain almost identical, the median aerosol number concentration must have decreased (see Figure A2). Regarding the sources of the aerosol particles at WOP during the night





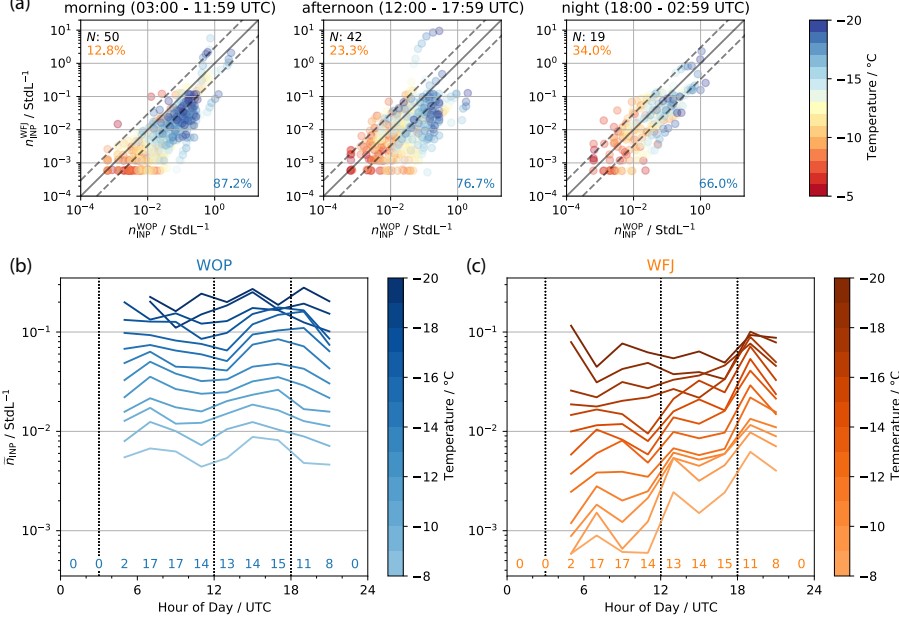

**Figure 4.** (a) Scatter plots of INP samples taken *in parallel* on the mountaintop (WFJ) and high valley (WOP) site in the morning, afternoon and night. Samples were considered to be *in parallel* when their sample time differed by less than 30 minutes. The color indicates the activation temperature of a given pair of INP concentrations. The solid line represents the 1:1 line whereas the dashed lines indicate an offset by a factor of three. The number of available samples ($N$) per period is indicated on the upper left. The percentage of observed INP concentrations at WFJ exceeding INP concentrations found at WOP and vice versa are given in the upper left and lower right respectively. (b) and (c) Time series of median INP concentration $\widetilde{n}_{\mathrm{INP}}$ at different temperatures (from $-8$ °C to $-20$ °C in steps of one K) for samples taken at WOP (b) and WFJ (c). The samples were binned in two hour intervals for the median calculations. The number of samples available within each two hour bin is indicated at the bottom of the plot. The dotted vertical lines represent the separation times used for data selection and to calculate median values across INP spectra (see Figure 5).

(i.e. between 0 UTC and 6 UTC) there are two likely pathways: ice-nucleation-inactive aerosol could (i) have been brought down with air masses from higher altitude (if INPs have been previously removed by solid precipitation) due to the nocturnal stratification of the atmosphere, or (ii) be emitted by local sources in the valley. For the latter, two potential sources are imaginable: First, soot emissions from traffic could be confined to the valley floor by a nighttime inversion layer. Fresh soot is not expected to contribute to the observed INP concentrations and thus reduce the $\mathrm{AF}_{0.5}$ (following the same argumentation as

in Section 3.1). Second, aerosol from biomass burning from heating of neighboring houses could also be confined in the valley. From the study of McCluskey et al. (2014), expected median $\mathrm{AF}_{0.5}$ associated with different biomass burning events range between $10^{-5}$ and $10^{-4}$ for temperatures of $-15$ °C and $-20$ °C, respectively. Consequently, biomass burning emissions containing hardly any INP would dilute the INP concentration within the total aerosol population, and thus decrease the $\mathrm{AF}_{0.5}$ at WOP. All three scenarios could explain the lower $\mathrm{AF}_{0.5}$ at WOP during the morning period, but remain speculative.



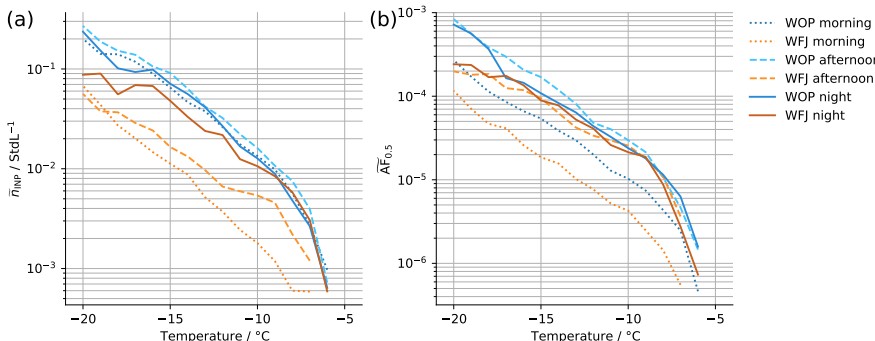

**Figure 5.** Median (cumulative) INP concentrations $\widetilde{n}_{\mathrm{INP}}$ (a) and median activated fractions $\widetilde{\mathrm{AF}}_{0.5}$ (b) for mountaintop (WFJ, orange) and high valley (WOP, blue) site for the three time periods of the day: Morning period (03:00 - 11:59 UTC) in dots, afternoon period (12:00 - 17:59 UTC) in dash and night period (18:00 - 02:59 UTC) in solid (see dotted lines in Figure 4b and 4c).

The decrease in median $\mathrm{AF}_{0.5}$ from the afternoon to the night at WOP is comparable to the decrease observed in median INP concentrations, indicating either an increase in ice-nucleation-inactive aerosol at WOP or depletion of INPs. At WFJ, the median $\mathrm{AF}_{0.5}$ increases proportionally more than the median INP concentrations from the morning to the afternoon, suggesting an increase of INPs or a decrease of ice-nucleation-inactive aerosol. Towards the night, the median $\mathrm{AF}_{0.5}$ at WFJ remains at a similar level as in the afternoon, while the median INP concentrations increased further. As local INP sources at WFJ are unlikely (cf. Section 3.1), the diurnal trend at WFJ indicates that air masses of same $\mathrm{AF}_{0.5}$ are likely to have mixed with the air masses at WFJ or replaced the air masses of lower INP and aerosol concentration at WFJ. Whether these air masses originated from the valley and were transported to WFJ or if WFJ and WOP are affected by the same air masses is not clear yet.

WFJ is susceptible to external perturbation (Section 3.1) and since the increase in INPs during the afternoon occurs on a daily basis, it is conceivable that it is caused by local or regional conditions rather than a global long-range transport phenomenon. A possible explanation would be a local or regional wind system by which INP are transported from lower altitude towards the mountaintop. That the valley serves as supply for INPs at higher altitude or is affected by the same air masses is supported by the equilibration of INP concentrations and activated fractions at both sites during the night.

### 3.3 Overview of the local topography and valley winds

In this section, potential transport mechanisms over orographic terrain are considered. For the subsequent analysis we divided the data into four meso-scale wind direction sectors (see Figure 1), each featuring a distinct topography. Towards the south and the west, the mountain ridges branching off WFJ were used as topographic divisions. For north and east, division lines were set orthogonal to the valley axes. Towards northwest (NW) the Prättigau valley stretches down to the Rhine valley (approx. 500 m a.s.l., see Figure 1). During NW wind, the channeled winds in the valley could transport aerosol and thus INPs from the Rhine valley towards WFJ and WOP (Figure 6a). From southwest (SW) the topography also features a valley stretching down

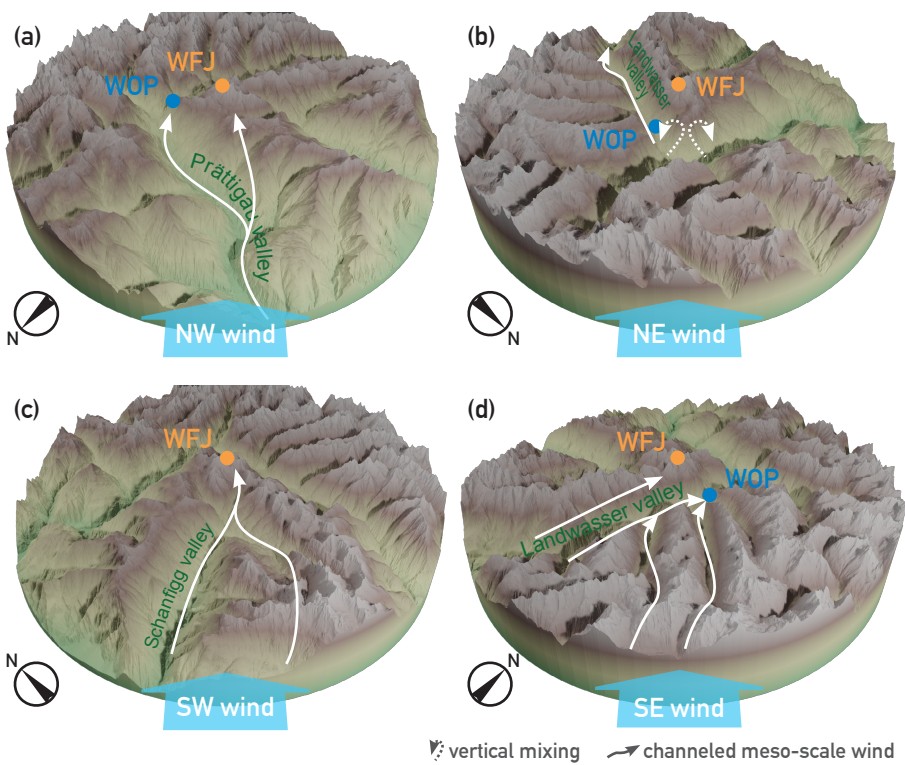

**Figure 6.** Channeled winds in the valleys (solid white arrows) of the meso-scale winds (light blue arrows) for the topography around the two measurement sites (WFJ: orange dot, WOP: blue dot) within a radius of 20 km around WFJ for the four wind direction sectors (as defined in Figure 1). The viewpoint in each panel is aligned with the meso-scale wind direction, is centered on WFJ (2693 m a.s.l.), and reaches down to an elevation of 500 m a.s.l.. In the NE wind case (b), the potential vertical mixing due to conditional instability of the narrow cross valley air masses is shown (dashed white arrows). The elevation data was obtained from the digital height model DHM2 from the Federal Office of Topography swisstopo.

to the Rhine valley (Figure 6c). Different to the NW case, the Schanfigg valley directly connects only to WFJ, while WOP is located leeward of WFJ. Thus, during SW wind, it is conceivable that we did not always observe the same air masses at WFJ and WOP. Same air masses are only observed if the air in the Landwasser valley is weakly stratified such that the air masses arriving at WFJ from the Schanfigg valley would follow the leeward slope down to WOP. The topography from southeast (SE) is more complex and the valley wind direction is sensitive to even small deviations in meso-scale wind direction (Figure 270    6d). When the wind direction is from the SE, near-surface winds are channeled within valleys in meso-scale flow direction. At the end, valley air masses would be deflected towards WOP by the mountain barrier around WFJ. Due to the inertia of the air masses they could reach WFJ if the wind speed is high enough and the stability of the air mass is favorable (i.e. the absence of a stable air mass layer potentially preventing rising air masses). If the wind direction is more southerly, however, near-surface winds would follow the Landwasser valley (cf. Figure 1) reaching WOP but not necessarily WFJ. Lastly, the


topography northeast (NE) of WFJ does not channel the flow upstream of WFJ as NE wind first passes over high mountain
ridges and then crosses the junction of the Prättigau and Landwasser valleys (Figure 6b). A north wind in the Landwasser valley
will be induced, but aerosol transport to WFJ might only be possible if the lower air masses are conditional unstably layered.
The Prättigau and Schanfigg valleys (see Figure 1) have the potential to act as ramps introducing INP-rich aerosol from lower
elevations to higher altitudes in the case of NW and SW meso-scale wind, respectively (Figure 6a and 6c). In the next section,
we further investigate this mechanism utilizing the Froude number, which can be used as a proxy for whether a low level air
mass is able to pass over mountain barriers.

### 3.4  Assessing air mass transport using the Froude number

The regime for air mass transport towards and over a mountain barrier can be estimated by the Froude number $Fr$ defined as
the ratio,

$$Fr = \frac{U_\infty}{h \cdot N} \tag{4}$$

of the synoptic wind speed $U_\infty$ to the barrier height $h$ times the Brunt-Väisälä frequency $N$. For $Fr \ll 1$ the flow is blocked and
transport over the barrier is hampered. If $Fr > 1$ air masses are transported over the barrier. This means that with increasing
$Fr$ transport of air masses from upstream of the barrier to downstream is promoted. In Figure 7 we present the WFJ INP
concentration at $-12$ °C as a function of $Fr$ for the four wind direction sectors. For the calculation of $Fr$ the wind speed
measured at WFJ was used, which was found to be representative for the meso-scale flow. The Brunt-Väisälä frequency was
calculated using the meteorological data from WFJ and an upstream weather station in each sector (RAG for NW, ARO for
SW, DAV for SE and NE, see Figure 1). For this approach we made the assumption that using data from two weather stations
located up to 30 km apart (distance WFJ-RAG) results in representative vertical gradients of pressure, relative humidity,
and temperature. We could validate this assumption for the NW wind direction sector using meteorological data from the
MeteoSchweiz weather station at Zurich airport (Kloten, 115 km NW of WFJ) as the lower reference point. The calculated
Brunt-Väisälä frequencies using the data from WFJ-RAG and WFJ-Zurich airport resulted in similar values (significant Pearson
correlation coefficient $r = 0.84$) corroborating our understanding of air masses being pushed both from the Swiss plain over
the Alps as a whole and locally from the Rhine valley to WFJ. It is clear that vertical transport will be suppressed in the
presence of an inversion layer, however as our measurements were performed during cloud events we attribute the likelihood
of such layers over the measurement period(s) as minor. For the cases of meso-scale NW and SW wind (Figure 7a and 7c), we
observed a significant moderate relation between INP concentration at -12 °C and $Fr$, in contrast to the NE and SE directions
(Figure 7b and 7d). This observation was indeed similar for INP concentrations at all temperatures between -8 °C and -16 °C
(see Table A1). We attribute the positive relation between INP concentration and $Fr$ to the NW and SW topography, such that
orographic lifting of air masses from the Rhine valley to WFJ occurs. Despite the absence of a significant or strong relation for
the SE case, one observation can still be made. With increasing INP concentration at WFJ, the average INP ratio (average ratio
between available INP concentrations between -10 °C and -14 °C at WFJ and WOP, see Equation 5) increases. The increase in
average INP ratio indicates that during meso-scale SE wind, air masses always reach WOP (i.e. INP ratio being independent



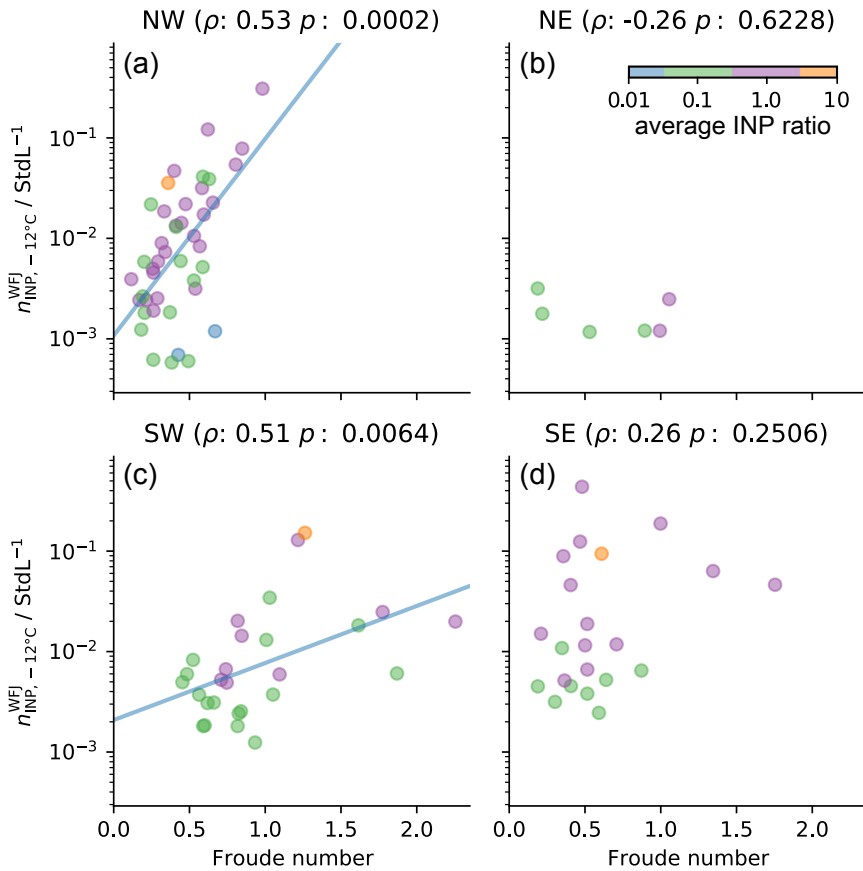

**Figure 7.** INP concentration at -12 °C at WFJ ($n_{\mathrm{INP},-12°C}^{\mathrm{WFJ}}$) as a function of Froude number (see Equation 4) for the four meso-scale wind direction sectors (as indicated in Figure 1). For each subplot the Spearman's rank coefficient ($\rho$) and the corresponding two-sided p-value ($p$) are shown at the top. The colors of the dots indicate the average INP ratio between WFJ and WOP (to assess the similarity of the INP concentration of both sites) as defined in Equation 5. Lines are exponential fits in case $|\rho| \geq 0.5$.





of the Froude number but proportional to the INP concentration at WFJ) and, to a varying degree, WFJ (Figure 7d). In the NW and SW cases, higher average INP ratios were also often found to coincide with higher INP concentration at WFJ. Thus, on

average INP concentrations at WFJ were rarely higher than at WOP and the increase in INP concentration at WFJ is larger than on WOP during air mass transport from NW and SW. This further supports the conclusion that the INP concentration at WFJ is more susceptible to aerosol perturbation. In the next section we will discuss the mixing of air masses between WFJ and WOP for all wind direction sectors in more detail.

### 3.5 Vertical mixing of air masses

To assess the vertical stability and mixing of the lower troposphere in the absence of clouds, the gradient of the potential temperature $\theta$ is typically utilized. It accounts for decreasing atmospheric pressure with increasing altitude, resulting in decreasing air density. Air masses are stably layered if $\theta$ increases with altitude (see e.g. Stull, 1988). In the opposite case of unstable layering, air masses with higher $\theta$ rise until the negative $\theta$ gradient diminishes. Our measurements specifically targeted cloud events. Thus, to assess vertical stability we utilized the equivalent potential temperature $\theta_e$, which in addition to pressure also

considers the latent heat that was released during condensation (Stull, 1988). To relate the INP concentrations measured at WFJ to WOP we calculate a ratio between two samples collected synchronously at both sites. The ratio in INP concentration $r_{\mathrm{INP},T}$ at a given temperature $T$ might not be available at all temperatures for a pair of samples (one from WOP, one from WFJ), as the overlap of available INP concentration varies between the samples. For example, if WOP experienced very ice-active air masses but not WFJ, INP concentrations at WOP could only be available down to -11 °C whereas at WFJ INP concentrations could not be available above -10 °C. Thus, $r_{\mathrm{INP},T}$ would only be available at -10 °C and -11 °C. In other cases $r_{\mathrm{INP},T}$ could

only be available at -14 °C. Thus, to generally assess the similarity of the observed INP concentration we take the mean of all available $r_{\mathrm{INP},T}$ between -10 °C and -14 °C, which is the temperature range where INP concentrations at WOP and WFJ overlapped most (see Figure 3). The *average INP ratio* was calculated as

$$\overline{r}_{\mathrm{INP}} = \mathrm{mean}\{r_{\mathrm{INP},m}\} = \frac{1}{|\mathcal{M}|} \sum_{m \in \mathcal{M}} \frac{n_{\mathrm{INP}}^{\mathrm{WFJ}}(m)}{n_{\mathrm{INP}}^{\mathrm{WOP}}(m)} \tag{5}$$

with $\mathcal{M}$ being the set of all temperatures between -10 °C and -14 °C at which an INP concentration $n_{\mathrm{INP}}^{\mathrm{WFJ}}(m)$ and $n_{\mathrm{INP}}^{\mathrm{WOP}}(m)$ was detectable at WFJ and WOP, respectively. In Figure 8 we present the average INP ratio between WFJ and WOP per wind direction sector (see Figure 1) as a function of $\theta_e$ gradient with height. An upstream weather station in each wind direction sector was used to calculate the $\theta_e$ gradient (SRS for NW, ARO for SW, DAV for SE and NE, see Figure 1).

In the previous section we presented evidence for advective transport of INP-rich aerosol from lower elevation to WFJ and

WOP during a period with meso-scale westerly wind. For meso-scale NW wind we observed a significant moderate correlation between the average INP ratio and the $\theta_e$ gradient (Figure 8a), i.e. more similar INP concentrations coinciding with higher vertical stability. This finding renders local mixing of air masses (e.g. by convection) unlikely, but confirms advection of the same air mass to the two sites as the underlying transport mechanism. Despite similar topography, we did not observe this trend to the same extent for meso-scale SW wind (Figure 8c). We attribute this to the fact that for this wind direction WOP is situated

leeward of the mountain ridge around WFJ, and thus the same air mass was not necessarily sampled at both sites. In the case

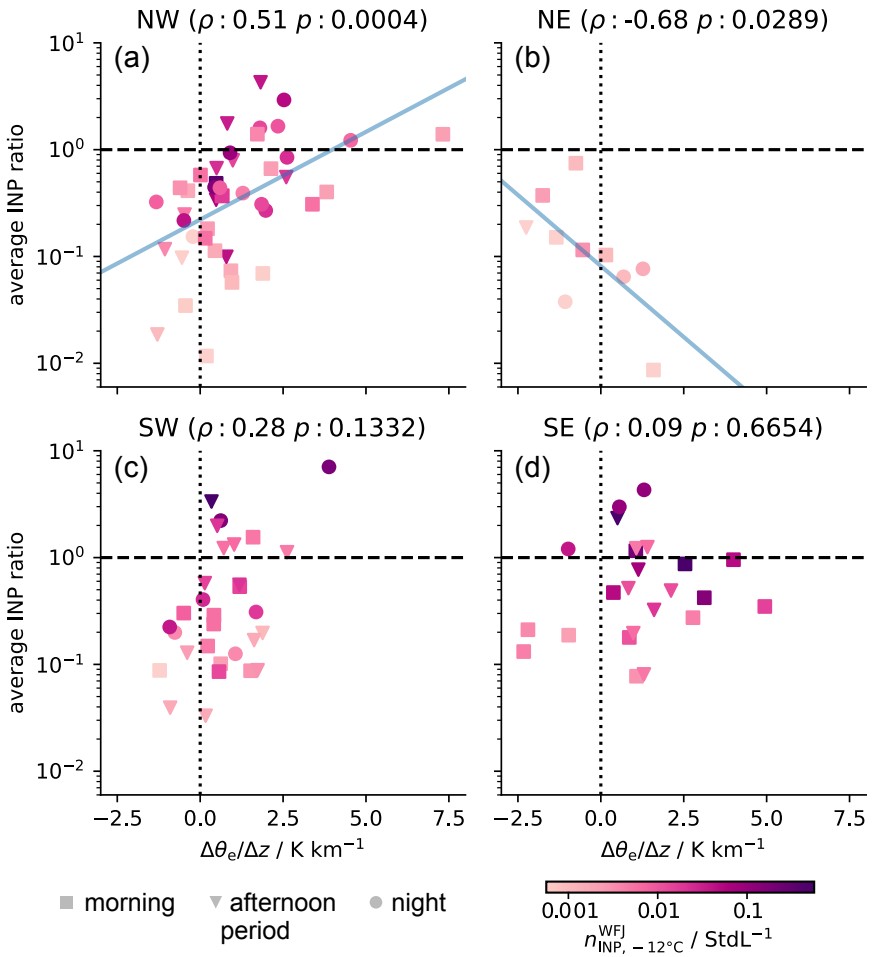

**Figure 8.** Average INP ratio ($\bar{r}_{INP}$, as defined in Equation 5) between synchronous INP samples taken at WFJ and WOP as a function of altitudal $\theta_e$ gradient for each wind direction sector. The dashed (horizontal) line indicates an average INP ratio of one. The dotted (vertical) line indicates $\theta_e$ gradient of zero representing a neutrally layered lower atmosphere. For each sector the Spearman's rank coefficient ($\rho$) and the corresponding two-sided p-value ($p$) is shown at the top. The marker shape indicates the sampling time (morning, afternoon, night, according to the time periods defined in Figure 4). The color of the markers indicates the INP concentration at -12 °C measured at WFJ. Blue solid lines are exponential fits in case $|\rho| \geq 0.5$.

of meso-scale NE wind, we observed increasingly coupled air masses between WFJ and WOP with decreasing $\theta_e$ gradient. This indicates that for the NE topography, air masses from the valley can reach WFJ when the lower air masses are conditional unstably layered (see Figure 6b). Strong convection as a transport mechanism seems unlikely in the wintertime Alps due to the extensive snow cover. However, a faster heating of valley air masses was observed (possibly due to a lower surface albedo





by vegetation, such as conifers) for the days being analyzed that could lead locally to slight instability. In the case of SE wind (Figure 8d) no relation between the two variables was found, such that no specific transport mechanism is discussed.

Previously we presented the diurnal cycle of median INP concentrations at WFJ (see Figure 5). In Figure 8 it can be observed that high average INP ratios were not exclusively observed during the night. Advection, which is most likely the underlying transport mechanism of INPs in the NW and SW cases, is not necessarily strongest towards the end of the day, but would vary

depending on the meso-scale flow. However, the distribution of the average INP ratios per period for the NW and SW cases exhibits the trend of progressively more similar INP concentrations at both sites over the course of day, which is resembled in the diurnal cycle at WFJ (Figure 5). Apart from a strong enough advection itself, the INP concentrations at WFJ likely depend on the air mass history and their potential INP uptake upstream of the site. In the next section we will examine the locations of potential INP uptake by the air masses arriving at WFJ.

**3.6  Connecting transport mechanisms to observed diurnal cycle**

For all samples considered in this study, the majority (67%) were collected during synoptic westerly wind situations (NW and SW cases combined, see Figure 7a and 7c) during which orographic lifting is the dominant transport mechanism of INPs towards WFJ. Thus, the diurnal cycle observed at WFJ (Figure 4) is likely a result of three phenomena that are superimposed to varying degree over the course of a day. These are: (i) the strength of advection of an air mass towards WFJ, (ii) the residual

time of an air mass close to INP sources, and (iii) the strength of INP sources at the time the air mass passed. In the previous sections we presented that in situations of westerly wind higher INP concentrations at WFJ coincided with signs of stronger advection. To assess the locations of potential sources and thus the potential uptake of INPs by air masses measured at WFJ, we present INP footprint maps on a regional scale in Figure 9 and on a global scale in Figure A3. Locations of potential INP sources are indicated with a circle where the ten minute time steps of a back-trajectory was less than 500 m above ground and and only

for altitudes less than 2500 m a.s.l., which we assume to be relevant for INP uptake. On a global scale we could not identify preferred INP source locations of air masses between the three time periods (Figure A3). Thus, we constrain our further analysis to potential source locations on the regional scale. Back-trajectories related to INP measurements in the morning (Figure 9a) passed frequently over the Swiss plateau and generally low INP concentrations were measured at WFJ. For INP measurements in the afternoon (Figure 9b), low trajectory heights above ground were found in rather close proximity to WFJ. Thus, the

increasing INP concentrations during this time can be attributed to local sources from the surrounding valleys. Following Huang et al. (2021) and Kanji et al. (2017), potential sources of INP active above -20 °C are bioaerosols (bacteria, fungi, leaf litter), lifted soil dust, and pollen emissions, as the lower-lying land was not fully snow-covered at the time of measurement. Hazel and alder, and starting in March also poplar, ash, and birch are common in the Davos region (MeteoSchweiz, 2021a) and even more so in the Rhine valley (MeteoSchweiz, 2021b). Anthropogenic activities such as biomass burning, industrial

processes and transportation also contributed to the aerosol population but are not thought to contribute to the increased INP concentration and activated fraction as discussed in Section 3.2. In addition, the local boundary layer height already increased by noon such that aerosol was brought to higher altitudes over the Rhine valley, increasing the minimal height for aerosol uptake by a passing air mass. For INP measurements during the night (Figure 9c), trajectories originated again over the Swiss



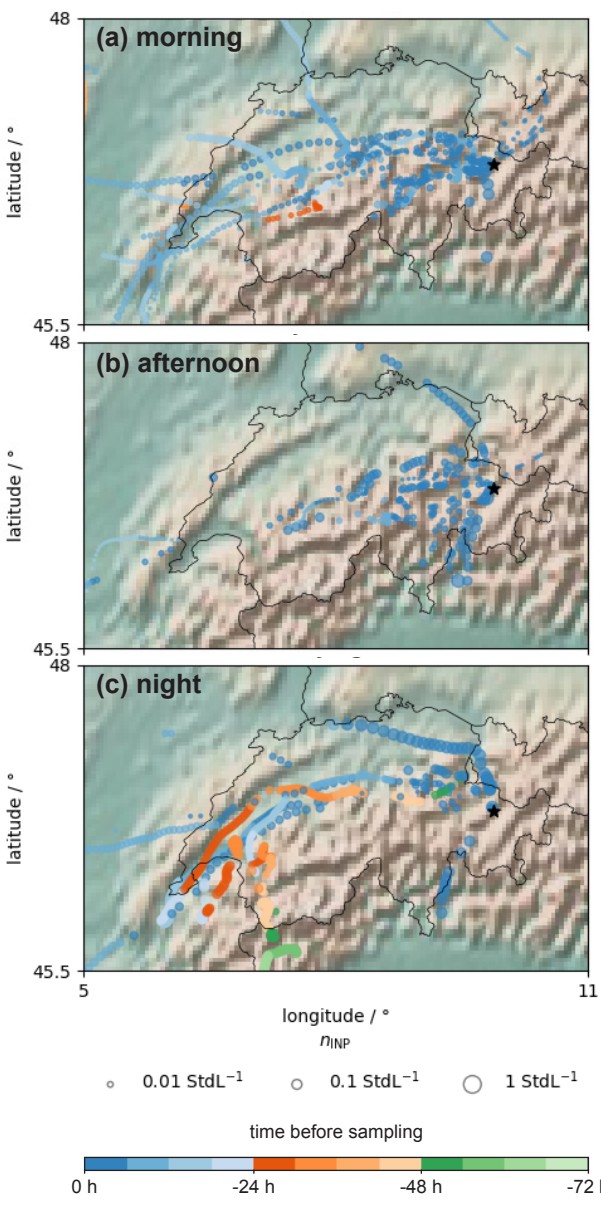

**Figure 9.** Footprint maps of potential INP uptake locations derived from back-trajectories associated with INP samples (one trajectory per sample) taken at WFJ for the periods morning (a), afternoon (b), and night (c). The orange circles indicate locations where a trajectory's ten minute time step was lower than 500 m above ground (and the surface height less than 2500 m a.s.l.). The size of a circle is proportional to the INP concentration at -12 °C measured at WFJ. The black star indicates the location of WFJ. The shaded relief was plotted with the python basemap library.





plateau and high INP concentrations were observed. In contrast to the morning, air masses moved slower (smaller distances
between individual circles in Figure 9c) and thus spent more time at low height over the Swiss plateau. Consequently, more
aerosol from the PBL over the Swiss plateau could be taken up by the air masses given the longer residence time. Due to
the lower altitude and larger agricultural areas, the contributions of bioaerosol and soil dust, respectively, can be assumed to
be stronger. In addition, the pollen season had already begun (MeteoSchweiz, 2021c) in January. From this we conclude that
during synoptic west wind, an increasing concentration of INPs was transported by advection towards WFJ over the course
of a day. While their origin in the beginning of the day was more likely to be within the closer proximity upstream of WFJ,
aerosol sourced in the Swiss plateau could have also contributed to the observed INP concentrations in the evening and night.

## 4 Conclusions

In this study we investigated the spatiotemporal distribution of INPs over the Swiss Alps near Davos in February and March
2019. Our analysis was based on simultaneous in situ INP measurements between 0 °C to −20 °C using offline drop-freezing
techniques on site and aerosol size distribution measurements in a high valley and on a nearby mountaintop. We used the vertical
gradient in equivalent potential temperature and the Froude number around the measurement sites to identify atmospheric
mixing and transport process regimes, respectively. Source regions of advective transport were suggested based on back-
trajectory footprint maps. After studying the diurnal variations in INP concentration at both sites during cloud events we
conclude the following:

– The median INP concentrations measured throughout the field campaign in February and March 2019 on Weissfluhjoch
  (WFJ) were lower than in the high valley at Wolfgangpass (WOP) by a factor of approximately three across the temper-
  ature spectrum. The distributions of INP concentrations measured at WFJ show larger variability which, together with
  the lower median INP concentrations, make WFJ more susceptible to perturbations of INP (local sources or long-range
  transported).

– The median INP concentrations at WFJ increased by nearly one order of magnitude over the course of a day, equilibrating
  to the concentrations measured at WOP, where median INP concentrations did not substantially change over the course
  of a day.

– We found significant correlations between the Froude number and the INP concentrations for periods with meso-scale
  wind directions from NW and SW, which indicates a transport of low level valley air masses to the mountain top site by
  forced orographic lifting. This finding could be generally true for a topography if the meso-scale wind direction aligns
  to the valley axis.

– We deduced that in the situation of a meso-scale wind perpendicular to a mountain valley, INP concentration on the
  mountaintop downstream can increase if the valley air masses are (conditional) unstable. The extent of this mechanism
  could not be shown with full significance, due to a low number of available samples with these conditions.





- 410    – The dominant transport mechanism across all observations was advection followed by orographic lifting. The resulting response in INP concentration observed on the mountaintop depends on (i) the strength of advection of air masses, and (ii) the upstream residual time of the air masses close within the PBL at low altitude. We conclude that INP-rich air masses are being pushed from the upstream valley and plains to the mountain ridge over the course of a day (Figure 6). Whereas the advected INPs originate from the surrounding valleys during the day, they source from the Swiss plateau

- 415    towards the night.

Additionally, we note the variability of the observed activated fraction, i.e. the absence of a relation between INP concentration and the aerosol (number) concentration. It implies that predicting continental INP concentrations at warmer temperatures ($T \geq -20$ °C) based on aerosol number concentration alone can be uncertain and that dynamics play the dominant role, especially over orographic terrain. Our study suggests that over orographic terrain and during strong mesoscale winds over the

Alps, mountaintop INP concentrations adjust to the INP concentrations of the valleys upstream over the course of a day. Thus, scaling existing INP paramterisations by an advective transport variable such as the Froude number used in this study, could lead to a more accurate representation of INP in model applications. To study INP transport mechanisms, gathering high resolution datasets of INP concentrations is of crucial importance. Despite the labor-intensive work using offline methods, this could be done at two nearby sites in this study. Recent developments gave rise to different autonomous online INP counters

(Bi et al., 2019; Brunner and Kanji, 2021; Möhler et al., 2021). Installing several units of these instruments in regions of interest can be used to improve the understanding of INP transport mechanisms in future studies. At the moment the minimal detectable concentrations and maximal operation temperatures are not suitable to study INPs are warm temperatures ($T > -25$ °C, Brunner and Kanji, 2021), but a next generation of online counters may overcome these limitations.

*Code and data availability.* Code is available on request. The data used in this study is publicly accessible on the campaign's website

https://www.envidat.ch/group/raclets-field-campaign.





## Appendix A

### A1    INP concentrations at WFJ per wind direction

To assess the potential influence of the restaurant west of the sampling site of WFJ, we show INP concentrations at different temperatures as a function of wind direction in Figure A1. An influence from the cooking for lunch would be expected from the west and between 10 am to 1 pm local time. No significant increase in INP concentration matching the two criteria was observed rendering a contribution from cooking emissions minuscule.

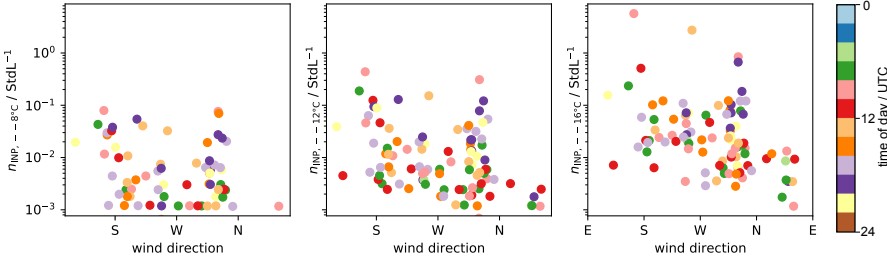

**Figure A1.** INP concentrations at -8 °C, -12 °C and -16 °C measured at WFJ as a function of wind direction. The color indicates the daytime of sampling.



## A2 Aerosol concentrations during investigated periods

Violin plots of the aerosol number concentrations with physical diameter larger than 0.5 μm during sampling for INP analysis at both sites for the morning, afternoon and night period (see Section 2.1.1) are shown in Figure A2. While the median
concentration at WFJ is relatively stable, it decreases at WOP by roughly a factor of five from the morning to the night. In addition, also the shape of the distributions at WFJ and WOP becomes similar at night indicating a better mixing of the atmosphere between high valley and mountaintop. We validated that the difference in aerosol number concentration between individual samples measured in parallel at WFJ and WOP indeed becomes smallest during the night.

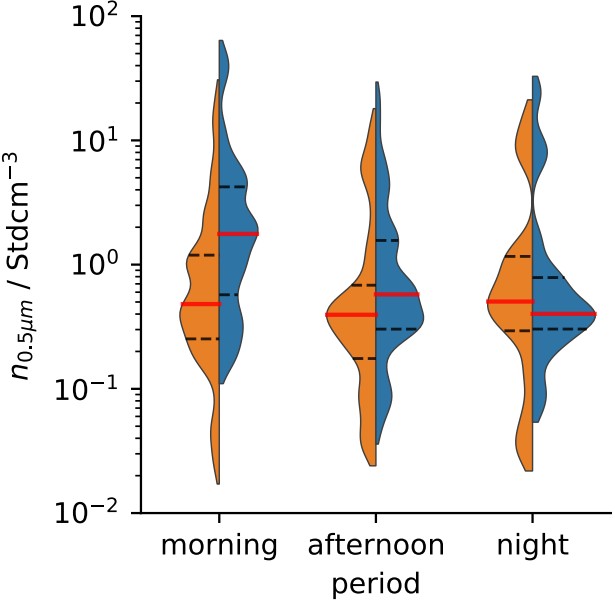

**Figure A2.** Violin plots of aerosol number concentrations for particles with physical diameter larger than 0.5 μm at WFJ (orange) and WOP (blue) in the morning (03:00 - 11:59 UTC), the afternoon (12:00 - 17:59 UTC) and night (18:00 - 02:59 UTC). Only measurements that coincide with INP samples at each side have been included.





## A3 Correlation coefficients of WFJ INP concentration at different temperature versus Froude number

The transport of INP towards WFJ was assessed using the Froude number. Table A1 summarizes Spearman's rank coefficients ($\rho$) between Froude number and WFJ INP concentrations at temperatures between -7 °C to -18 °C (in the main text only the results for INP concentrations at -12 °C were shown). A continuous range of significant relations was found for synoptic NW and SW wind (see Figure 1) between -8 °C to -16 °C.

**Table A1.** Spearman's rank coefficients between Froude number and INP concentrations at different temperatures (-7 °C to -18 °C) per wind sector (see Figure 1). Numbers in bold represent a significant result (two-sided $p < 0.05$). At -7 °C only two INP concentrations were available at NE such that no Spearman's rank coefficient could be calculated.

| $\rho$ | -7 °C | -8 °C | -9 °C | -10 °C | -11 °C | -12 °C | -13 °C | -14 °C | -15 °C | -16 °C | -17 °C | -18 °C |
|---|---|---|---|---|---|---|---|---|---|---|---|---|
| NW | **0.56** | **0.58** | **0.57** | **0.61** | **0.55** | **0.53** | **0.47** | **0.48** | **0.41** | **0.48** | 0.35 | 0.32 |
| NE | | 0.4 | 0.4 | **0.9** | 0.26 | -0.26 | -0.17 | 0.08 | 0.33 | 0.1 | 0.47 | 0.38 |
| SE | 0.3 | 0.28 | 0.33 | 0.33 | 0.37 | 0.26 | 0.29 | 0.3 | 0.19 | 0.13 | 0.04 | -0.27 |
| SW | 0.32 | **0.55** | **0.56** | **0.55** | **0.55** | **0.51** | **0.47** | **0.46** | **0.58** | **0.57** | **0.5** | **0.47** |

## A4 Diurnal global back-trajectory maps

Figure A3 shows back-trajectories associated with all samples at WFJ for the morning, afternoon, and night period (see Section 2.1.1). No preferential source region could be identified on a global scale.

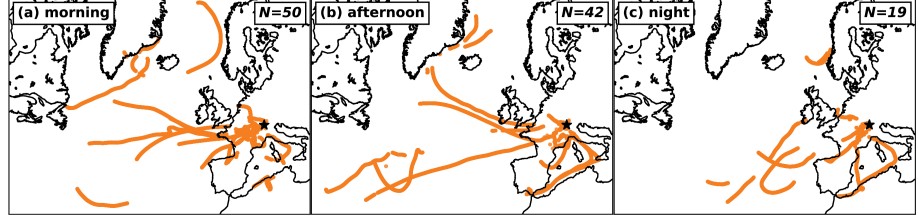

**Figure A3.** Footprint map calculated from back trajectories of each sample taken at WFJ for the periods morning (a), afternoon (b) and night (c) as defined in Sec. 2.1.1. The dots indicate places where the trajectories were lower than 500 m above ground. The number of trajectories per plot ($N$) is listed in the upper right corner of each plot. The black star indicates the location of WFJ.



*Author contributions.* JW performed the analysis and prepared the figures for the manuscript. JW and CM installed and maintained the aerosol instruments in Davos. JW, CM, MSc and LR performed the aerosol measurements during the campaign. MSp provided the back-trajectory data and helped with their analysis. CB processed the topographic data from the DHM2 model. JW, CM, JH, UL, CB and ZAK interpreted the data. JW wrote the manuscript with contributions from all co-authors. All authors reviewed the manuscript. ZAK supervised the project.

*Competing interests.* The authors declare no conflict of interest.

*Acknowledgements.* The authors express their gratitude to the RACLETS campaign team for their technical support and many fruitful discussions. Especially, we thank Michael Lehning (WSL/SLF, EPFL) and his team for their support in realizing the RACLETS campaign. We thank Paul Fopp for providing his land at Wolfgangpass and Martin Genter for logistical support at Weissfluhjoch. Our deepest appreciation to Michael Rösch and Marco Vecellio for technical support. We express our deepest gratitude to Nora Els (University of Innsbruck) for providing us with a second impinger. We thank MeteoSchweiz for the meteorological observations. The authors thank Franz Conen, Robert O. David, Alexandra Moniz, Fabiola Ramelli, Carolin Rösch, Maxim Samarin and Colin Tully for discussions and suggestions improving the manuscript. This study received funding from the Swiss National Science Foundation (grant numbers 200021_169620, 200021_175824).



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
