# Peer review of "Unveiling atmospheric transport and mixing mechanisms of ice nucleating particles over the Alps"

_Atmospheric Chemistry and Physics, 2021_

## Author Comment (AC1)

Reviewer comments are reproduced in **bold** and author responses in *italic*; extracts from the original manuscript are presented in *red italic*, and from the revised manuscript in *blue italic*.

**The authors present the INP measurements from two sites. They find that INP concentrations at the mountaintop site are on average lower than high valley site. The authors discuss the importance of the orographic effect towards the INP concentrations. This is a unique study where the authors have analyzed the field INP data in the context of meteorology. I have the following minor comments, and after addressing these comments I recommend publication.**

*We want to thank the anonymous referee for reviewing our manuscript. We are pleased with the positive reception and grateful for the helpful comments which improved our manuscript and are answered individually hereafter.*

**Line 13: Sentence needs to be revised. Maybe add Our results "show that" the local ....**

*We thank the reviewer for catching this typo. We changed line 13 (revised manuscript) as follows:*

*Our results* *suggest a* *local INP concentration enhancement over the Alps during cloud events.*

**Section 2.1: It is not clear why a heated inlet was used. Would this affect the composition of ambient aerosol? Do volatile components of these aerosol will be evaporated? On Line 110, can a special feature of the inlet design (that prevents snow sampling can be explained here? What is the cut size of this inlet?**

*We thank the reviewer and agree that more information is needed. We changed lines 109-118 (revised manuscript) as follows:*

*Similar to Weingartner et al. (1999), both* *inlets were capped with a hat* *preventing snow and while sampling particles with diameter smaller than 40 µm from entering the inlet for wind speeds of up to 20 m s$^{-1}$. All outside parts (including the hat, at WOP approx.* *the first 0.7 meters, at WFJ all* *parts) were heated to 46 °C to avoid riming on the outside parts, to sublimate ice crystals, and to evaporate activated cloud droplets. The evaporation of volatile compounds of the aerosol cannot be excluded. However, the relevant ice active particles in the investigated temperature regime (T ≥ -20 °C) are mostly biological which should only degrade at temperature higher than 46 °C (Kanji et al. 2017; Huang et al., 2021). In addition, the flow rate through the inlet is high (300 L min$^{-1}$), as such the aerosol flow was likely at temperatures below 46 °C at which INPs typically do not become inactive. Contributions of resuspended particles from the snow-covered surface around the measurement sites cannot fully be excluded but are unlikely to have added significantly to the sampled aerosol due to the inlet's design (Mignani et al., 2021).*

**Section 3.4: Equation 4, how height, h, is calculated?**

*We thank the reviewer for pointing out the need for a definition. We changed lines 307-310 (revised manuscript) as follows:*

*[…] which was found to be representative for the mesoscale flow. The difference in altitude of the subsequently described weather stations was used as barrier height (*h*) per wind sector. The Brunt-Väisälä frequency was calculated using the meteorological data from WFJ and* *the respective upstream weather station in each sector (RAG for NW, ARO for SW, DAV for SE and NE, see Figure 1).*

**Section 3.5: Did the vertical profiles of potential temperature were measured? In Figure 6 (line 262), it is mentioned: "the potential vertical mixing" – please elaborate. It is not clear whether vertical mixing occurred or not. If yes, how this is justified. It is not clear what test was used to confirm the vertical mixing.**

*We thank the reviewer for pointing out the need for more clarity. We changed the caption of Figure 6 as follows:*

*In the NE wind case (b), the potential vertical mixing (i.e. rising of air masses originating from the valley and mixing with the air masses aloft) due to conditional instability of the narrow cross valley air masses is shown (dashed white arrows).*

*In addition, we phrased the calculation of the $\theta_e$ more explicitly. We changed lines 350-353 (revised manuscript) as follows:*

*An upstream weather station in each wind direction sector was used to calculate $\theta_e$ at the valley floor (SRS for NW, ARO for SW, DAV for SE and NE, see Figure 1). The $\theta_e$ gradient was calculated by dividing the difference in $\theta_e$ between WFJ and the respective upstream weather station by the height difference of both stations.*

**Conclusions: To enhance the impact, how INP concentrations observed in this study compare with other field studies? Line 416, is this "absence of a relation" view supported by other studies? Can APS (Figure 2e) data be shown here? On line 418, it is said that this relation does not hold for a temperature warmer than -20 degC. Is this temperature threshold based on the present study? Are experiments are performed at colder temperatures to conclude this statement?**

*We thank the reviewer for the suggestion and agree with the necessity of a comprehensive comparison. To our best knowledge, Conen et al. 2017 is the only comparable study available. The same method (drop freezing assay) is used and the vertical distribution of INP in the Alpine region is investigated. Many studies measured INP at Jungfraujoch in the Bernese Alps. However, Jungfraujoch at 3580 m a.s.l. is located higher than WFJ and available studies at Jungfraujoch investigated INPs at lower temperatures and with different methods (e.g., Lacher et al. 2017; Brunner et al. 2021) or collected particles of larger size (Creamean et al. 2019). Hence, we compare to Conen et al. 2017 which is discussed in Section 3.1.*

*To support our argument for the absence of a relation between INP concentration and aerosol number concentration, we added Appendix A3 (see below) presenting correlation coefficients between the two observables. We added lines 264-268 (revised manuscript) to link the new appendix to the main text:*

[revised manuscript text omitted]

---

## Author Comment (AC2)

Reviewer comments are reproduced in **bold** and author responses in *italic*; extracts from the original manuscript are presented in *red italic*, and from the revised manuscript in *blue italic*.

**This is a careful detailed study of INP from two sites separated vertically over complex terrain. The vertical transport of INP is investigated. Overall, I suggest publication with the following minor revisions.**

*We want to thank the anonymous referee for reviewing our manuscript. We are pleased with the positive reception and grateful for the helpful comments which improved our manuscript and are answered individually hereafter.*

**Abstract Line 3: I would state this phrase more carefully. "INP, which measurements suggest are sparsely populated in the troposphere.**

*We thank the reviewer for the advice. We changed line 3 (revised manuscript) as proposed:*

*Primary ice crystals are formed on ice nucleating particles (INPs), which measurements suggest are sparsely populated in the troposphere.*

**Introduction Line 15: Field and Heymsfield, 2015 is not an appropriate or original reference for this statement.**

*We thank the reviewer and agree with the critique. Since the statement is very general, any publication would be more specific than appropriate. We therefore deleted the reference (line 16 in the revised manuscript).*

**Starting at Line 38. It is important to clarify earlier that an idealized PBL is in a well- mixed state with constant potential temperature and aerosol number concentration with height. This is stated later an idealized description, but should instead be clarified initially.**

*We thank the reviewer for pointing out the need for clarification. We changed lines 37-43 (revised manuscript) as follows:*

*Most atmospheric aerosol particles have their sources near Earth's surface and are therefore found within the first kilometers above ground in the troposphere, i.e. they are confined within the planetary boundary layer (PBL). In an idealized view, any PBL is in a well-mixed state when the potential temperature, aerosol number concentrations, and total humidity (sum of specific humidity and cloud water) are constant with height (see e.g. Stull, 1988; Chow et al., 2013). The top of a well-mixed PBL is, among other indicators, characterized by an abrupt increase in potential temperature, decrease in aerosol number concentration, and decrease in total humidity (Stull, 1988; Chow et al., 2013). In contrast to the description of a well-mixed PBL over a flat surface, an accurate description of the PBL over complex mountainous terrain is complicated by [...]*

**Line 56 – I am confused by term "arbitrary"**

*We thank the reviewer and agree that the phrasing can be confusing. We changed lines 56-58 (revised manuscript) as follows:*

*Any aerosol particle can potentially be removed by dry deposition or impaction scavenging. Additionally, aerosol particles can be removed by nucleation scavenging if embedded in hydrometeors after serving as INP or cloud condensation nuclei (Lohmann et al., 2016b).*

**Line 115 – more information is needed pertaining to the inlet. What is the efficiency of the inlet at different aerosol sizes?**

*We thank the reviewer and agree that more information is needed. Including the comment of referee 1 we changed lines 109-118 (revised manuscript) as follows:*

*Similar to Weingartner et al. (1999), both inlets were capped with a hat preventing snow and while sampling particles with diameter smaller than 40 µm from entering the inlet for wind speeds of up to 20 m s$^{-1}$. All outside parts (including the hat, at WOP approx. the first 0.7 meters, at WFJ all parts) were heated to 46 °C to avoid riming on the outside parts, to sublimate ice crystals, and to evaporate activated cloud droplets. The evaporation of volatile compounds of the aerosol cannot be excluded. However, the relevant ice active particles in the investigated temperature regime (T ≥ -20 °C) are mostly biological which should only degrade at temperature higher than 46 °C (Kanji et al. 2017; Huang et al., 2021). In addition, the flow rate through the inlet is high (300 L min$^{-1}$), as such the aerosol flow was likely at temperatures below 46 °C at which INPs typically do not become inactive. Contributions of resuspended particles from the snow-covered surface around the measurement sites cannot fully be excluded but are unlikely to have added significantly to the sampled aerosol due to the inlet's design (Mignani et al., 2021).*

**Line 190 – clarify why locations were omitted, since trajectories passed close to mountain tops? The reasoning here is unclear.**

*We thank the reviewer for pointing out the need for clarification. If this second criterion was not applied many individual points scattered across the Alpine region where high peaks were located. As from our understanding no considerable INP and aerosol sources are located on mountaintops we omitted these locations for easier interpretation of the plot. We changed lines 196-199 (revised manuscript) as follows:*

*[…] the trajectory was less than 500 m above ground indicating a potential aerosol source from these locations. Based on this criterion, also many locations near mountaintops over the Alps with heights above 2500 m a.s.l. were indicated as sources. As these locations are considered snow-covered in wintertime and thus feature no substantial sources of aerosol and INP, locations above the arbitrarily chosen surface height 2500 m a.s.l. were omitted.*

**Figure 3 – excellent plot, easy to understand**

*We thank the reviewer for the compliment.*

**Line 237 – this can also be caused by a combination of both (down from higher altitude and local sources)**

*We thank the reviewer and agree with the critique. We changed line 246 (revised manuscript) as follows:*

*Regarding the sources of the aerosol particles at WOP during the night (i.e. between 0 UTC and 6 UTC) there are two likely pathways potentially occurring simultaneously with varying magnitude: ice-nucleation-inactive aerosol could (i) […]*

**Line 244 – what 3 scenarios are referenced here? soot, biomass burning (with and without dilution), or... This may need to be rephrased.**

*We thank the reviewer for pointing out the need for clarification. We changed lines 255-256 (revised manuscript) as follows:*

*All three scenarios (sedimentation of aerosol from aloft, confined ground emissions from traffic and heating emissions) could explain the lower AF$_{0.5}$ at WOP during the morning period but remain speculative.*

**Figure 5, can you speak to the uncertainty of the measurement (or at least statistics around the medians).**

*We thank the reviewer and agree that more details are necessary. We adapted Figure 5 indicating significant changes in median INP concentration at WFJ and changed lines 239-262 (revised manuscript) as follows:*

*In Figure 5a and 5b the median INP concentration and the median activated fraction (AF$_{0.5}$, ratio of INP concentration and total aerosol concentration of particles with diameter larger 0.5 μm), respectively are shown as a function of temperature for the three time periods (morning, afternoon, night). The significance of differences of the medians at WFJ in the morning compared to in the afternoon and night, respectively, was determined using the Mann–Whitney U test (significance level p<0.05) per temperature. As there is a significant increase in median AF$_{0.5}$ at WOP from the morning to the afternoon, whereas the corresponding median INP concentration remains almost identical, the median aerosol number concentration must have decreased (see Figure A2). […] At WFJ, the median AF$_{0.5}$ (significantly) increases proportionally more than the median INP concentrations from the morning to the afternoon, suggesting an increase of INPs or a decrease of ice-nucleation-inactive aerosol. Towards the night, the median AF$_{0.5}$ at WFJ remains at a similar level as in the afternoon, while the median INP concentrations significantly increased further.*

[Figure]

**Figure 5.** *Median (cumulative) INP concentrations ñ$_{INP}$ (a) and median activated fractions ÃF$_{0.5}$ (b) for mountaintop (WFJ, orange) and high valley (WOP, blue) site for the three time periods of the day: Morning period (03:00 - 11:59 UTC) in dots, afternoon period (12:00 - 17:59) in dash and night period (18:00 - 02:59 UTC) in solid (see dotted lines in Figure 4b and 4c). Above panel (a), stars indicate if the median INP concentration at WFJ in the afternoon and night, respectively, is significantly (significance level p < 0.05) different from in the morning based on a Mann–Whitney U test per temperature.*

**Line 254 – Important point is that global long-range transport events were removed. This should be stated at other times within the paper.**

*We thank the reviewer for the advice. We changed lines 271 (revised manuscript) as follows:*

*WFJ is susceptible to external perturbation (Section 3.1) and since the increase in INPs during the afternoon occurs on a daily basis, it is conceivable that it is caused by local or regional conditions rather than global long-range transport (which is excluded based on the sample selection, see Section 2.1.1).*

*Additionally, we added lines 387-389 (revised manuscript):*

*On a global scale we could not identify preferred INP source locations of air masses for the three time periods (Figure A3). Note that INP contributions from long-range transport such as Saharan dust are unlikely due to the selection of the analyzed samples (see Section 2.1.1).*

**Figure 6 – can you add the % of cases (or number of cases) for each of the 4 wind scenarios?**

*We thank the reviewer for the advice. We changed Figure 6 as follows:*

[Figure]

*Figure 6.* *Channelled winds in the valleys (solid white arrows) of the mesoscale winds (light blue arrows) for the topography around the two measurement sites (WFJ: orange dot, WOP: blue dot) within a radius of 20 km around WFJ for the four wind direction sectors (as defined in Figure 1). The viewpoint in each panel is aligned with the mesoscale wind direction, is centered on WFJ (2693 m a.s.l.), and reaches down to an elevation of 500 m a.s.l.. The percentage of INP samples collected for each wind sector is indicated in the lower right of each subplot. In the NE wind case (b), the potential vertical mixing (i.e. rising of air masses originating from the valley and mixing with the air masses aloft) due to conditional instability of the narrow cross valley air masses is shown (dashed white arrows). The elevation data was obtained from the digital height model DHM2 from the Federal Office of Topography swisstopo.*

**Figure 7 – the use of "/" rather than (StdL^1) is quite confusing. It appears as a ratio rather than a unit.**

*We thank the reviewer for the comment, and we acknowledge that some readers could be used to a different style of unit notation. However, to stay consistent throughout the manuscript we prefer to retain the notation as is.*

**Figure 8 – are all 4 panels needed. There is very little science results outside of the NW plot. Could the other 3 panels be moved to supplemental figures?**

*We thank the reviewer for the suggestion. However, we think that valuable scientific insight can be gained from subplots 8b to 8d. Without Figure 8b the mixing of valley and mountain air masses could not be assessed (cf. Figure 7b). Figure 8c does not exhibit a strong relation between average INP ratio and potential equivalent temperature gradient (due to the location of WOP leeward of WFJ) but exhibits the same trend as the NW case supporting our hypothesis of orographic lifting. We acknowledge that the SE case does not allow to identify a specific transport mechanism. From our point it is important to still show this case as it illustrates that if the topography is too complex (e.g. no ramp-like as in the NW case) no relation to meteorological variable can be found. To underline this statement, we changed lines 365-368 (revised manuscript) as follows:*

*In the case of SE wind (Figure 8d) no relation between the two variables was found, such that no specific transport mechanism could be identified. The SE topography illustrates that an assessment of changes in INP concentration based on meteorological parameters requires a rather simple topography such as e.g. a ramp-like structure in the NW case.*

*Lastly, by keeping all subplots, the comparison between Figures 6, 7, and 8 is from our point of view more lucid as always all wind directions are presented.*

**Figure 9: I found this confusing. The color bar states that orange references 24-48 hours before sampling. Yet the caption states that orange circles indicate locations where a trajectory's ten minute time step was lower than 500 m above ground. Why are these two conditions met at the same time?**

*We thank the reviewer for catching this typo which was caused by a versioning error. The color of the circles indicated the time before sampling the corresponding air mass at WFJ. We changed the caption of Figure 9 as follows:*

*The circles indicate locations where a trajectory's ten-minute time step was lower than 500 m above ground (and the surface height less than 2500 m a.s.l.). The color of a circle indicates the time before the corresponding air mass was sampled at WFJ. The size of a circle is proportional to the INP concentration at -12 °C measured at WFJ.*

**Line 370-385 – This is speculative and potentially should be phrased with less certainty.**

*We thank the reviewer and agree that a less certain phrasing should be applied. We changed lines 393-411 (revised manuscript) as follows:*

*Thus, the increasing INP concentrations during this time could be attributed to local sources from the surrounding valleys. Following Huang et al. (2021) and Kanji et al. (2017), potential sources of INP active above -20 °C are bioaerosols (bacteria, fungi, leaf litter), lofted soil dust, and pollen emissions, of the lower-lying land, which was not fully snow-covered at the time of measurement. From February on, Hazel and alder are common in the Davos region (MeteoSchweiz, 2021a) and even more so in the Rhine valley (MeteoSchweiz, 2021b). Starting in March, poplar, ash, and birch also contribute to the pollen population. Anthropogenic activities such as biomass burning, industrial processes and transportation also contributed to the aerosol population but are not thought to contribute to the increased INP concentration and activated fraction as discussed in Section 3.2. In addition, the local boundary layer height could have already increased by noon such that aerosol could be brought to higher altitudes over the Rhine valley, potentially increasing the minimal height for aerosol uptake by a passing air mass. For INP measurements during the night (Figure 9c), trajectories originated again over the Swiss plateau and high INP concentrations were observed. In contrast to the morning, air masses moved slower (smaller distances between individual circles in Figure 9c) and thus spent more time at low height over the Swiss plateau. Consequently, more aerosol from the PBL over the Swiss plateau could be taken up by the air masses given the longer residence time at lower altitude. Due to the*

*lower altitude and larger agricultural areas, the contributions of bioaerosol and soil dust, respectively, can be assumed to be stronger. In addition, the pollen season had already begun (MeteoSchweiz, 2021c) in January. From this we conclude that during synoptic west wind, an increasing concentration of INPs was potentially transported by advection towards WFJ over the course of a day. While their origin in the beginning of the day was more likely to be within the closer proximity upstream of WFJ, aerosol sourced in the Swiss plateau could have also contributed to the observed INP concentrations in the evening and night.*

**References**

Huang, S., Hu, W., Chen, J., Wu, Z., Zhang, D., and Fu, P.: Overview of biological ice nucleating particles in the atmosphere, Environment International, 146, https://doi.org/10.1016/j.envint.2020.106197, 2021.

Kanji, Z. A., Ladino, L. A., Wex, H., Boose, Y., Burkert-Kohn, M., Cziczo, D. J., and Krämer, M.: Overview of Ice Nucleating Particles, Meteorological Monographs, 58, 1.1–1.33, https://doi.org/10.1175/AMSMONO-GRAPHS-D-16-0006.1, 2017.

Lohmann, U., Lüond, F., and Mahrt, F.: An Introduction to Clouds: From the Microscale to Climate, Cambridge Univ. Press, 2016b.

Mignani, C., Wieder, J., Sprenger, M. A., Kanji, Z. A., Henneberger, J., Alewell, C., and Conen, F.: Towards parameterising atmospheric concentrations of ice-nucleating particles active at moderate supercooling, Atmospheric Chemistry and Physics, 21, 657–664, https://doi.org/10.5194/acp-21-657-2021, 2021.

Weingartner, E., Nyeki, S., and Baltensperger, U.: Seasonal and diurnal variation of aerosol size distributions (10 < D < 750 nm) at a high-alpine site (Jungfraujoch 3580 m asl), Journal of Geophysical Research Atmospheres, 104, 26 809–26 820, https://doi.org/10.1029/1999JD900170, 1999.